# The Role of Virulence Factors in Neonatal Sepsis Caused by Enterobacterales: A Systematic Review

**DOI:** 10.3390/ijms231911930

**Published:** 2022-10-08

**Authors:** Lucia Barcellini, Giulia Ricci, Ilia Bresesti, Aurora Piazza, Francesco Comandatore, Mike Sharland, Gian Vincenzo Zuccotti, Laura Folgori

**Affiliations:** 1Department of Paediatrics, Children Hospital V. Buzzi, University of Milan, 20154 Milan, Italy; 2Division of Neonatology, “F. Del Ponte” Hospital, Woman and Child Department, University of Insubria, 21100 Varese, Italy; 3Unit of Microbiology and Clinical Microbiology, Department of Clinical-Surgical, Diagnostic and Pediatric Sciences, University of Pavia, 27100 Pavia, Italy; 4Department of Biomedical and Clinical Sciences, Romeo and Enrica Invernizzi Pediatric Clinical Research Center, University of Milan, 20100 Milan, Italy; 5Centre for Neonatal and Paediatric Infection, Institute for Infection and Immunity, St. George’s University of London, London SW17 9PE, UK

**Keywords:** neonatal sepsis, *Enterobacterales*, virulence factors, virulome, mortality

## Abstract

Neonatal sepsis is a life-threatening condition with high mortality. Virulence determinants relevant in causing Gram-negative (GN) neonatal sepsis are still poorly characterized. A better understanding of virulence factors (VFs) associated with GN neonatal sepsis could offer new targets for therapeutic interventions. The aim of this review was to assess the role of GN VFs in neonatal sepsis. We primarily aimed to investigate the main VFs leading to adverse outcome and second to evaluate VFs associated with increased invasiveness/pathogenicity in neonates. MEDLINE, Embase, and Cochrane Library were systematically searched for studies reporting data on the role of virulome/VFs in bloodstream infections caused by *Enterobacterales* among neonates and infants aged 0–90 days. Twenty studies fulfilled the inclusion criteria. Only 4 studies reported data on the association between pathogen virulence determinants and neonatal mortality, whereas 16 studies were included in the secondary analyses. The quality of reporting was suboptimal in the great majority of the published studies. No consistent association between virulence determinants and GN strains causing neonatal sepsis was identified. Considerable heterogeneity was found in terms of VFs analysed and reported, included population and microbiological methods, with the included studies often showing conflicting data. This variability hampered the comparison of the results. In conclusions, pathogens responsible for neonatal sepsis are widely heterogenous and can use different pathways to develop invasive disease. The recent genome-wide approach needs to include multicentre studies with larger sample sizes, analyses of VF gene profiles instead of single VF genes, alongside a comprehensive collection of clinical information. A better understanding of the roles of virulence genes in neonatal GN bacteraemia may offer new vaccine targets and new markers of highly virulent strains. This information can potentially be used for screening and preventive interventions as well as for new targets for anti-virulence antibiotic-sparing therapies.

## 1. Introduction

Neonatal sepsis is defined as a systemic condition of infectious origin associated with hemodynamic changes and other clinical manifestations leading to life-threatening organ dysfunction. Despite clinical progress, neonatal sepsis still causes substantial morbidity and mortality worldwide. A recent meta-analysis estimated the global incidence of neonatal sepsis to be 2824 per 100,000 live births [1], significantly higher than estimates reported in 2018 of 2202 per 100,000 live births [2]. The mortality rate reported was 17.6% [1]. In the last decades, efforts toward maternal intrapartum antimicrobial prophylaxis have significantly reduced the rates of Group B *Streptococcus* (GBS) disease but have been associated with increased rates of infections driven by Gram-negative bacteria (GNB). *Escherichia coli* already surpassed GBS as the most common cause of bloodstream infections (BSIs) in premature newborns, and it is the most common cause of mortality among all gestational ages [3,4]. Moreover, the epidemiology of neonatal sepsis varies across geographies, and recent studies have shown that in low- and middle-income countries (LMICs), GNB are the most frequently reported cause of bacterial neonatal sepsis, particularly *Klebsiella* spp. [5]. No vaccine or other preventive strategy is available to control neonatal GNB sepsis, and the increasing antimicrobial resistance rate among GNB are a public health concern.

Neonatal sepsis is classified based on the timing of presentation as early onset (EOS) or late onset (LOS). LOS is often considered a hospital-acquired infection in which major contributors to pathophysiology are the immature immune system, the wide exposure to environmental pathogens, and the presence of invasive devices. On the other hand, the pathogenesis of early-onset vertically transmitted GNB bacteraemia is not completely understood. *E. coli* typically colonizes the gastrointestinal tract of human infants within a few hours after birth with exposure to the mother’s microbiota and the environment. Only highly adapted clones that have acquired specific virulence attributes which confer on them the ability to survive in different environments/niches (such as blood) can cause broad spectrum diseases. Virulence determinants relevant in causing neonatal GNB sepsis are still poorly characterized, with most data derived from animal models of *E. coli* meningitis. Less is known on the pathophysiology of neonatal sepsis prior to or without meningitis. It has been shown experimentally that bacterial passage across the neonatal ingestion of maternal *E. coli* is an early step in neonatal sepsis pathogenesis [6]. Moreover, neonatal *E. coli* strains invade and transcytose intestinal epithelial cells [7]. Virulence factors (VFs) contributing to intestinal translocation or blood invasion/survival are likely to be needed for neonatal septicaemia prior to or in the absence of meningitis.

A better understanding of VFs associated with GNB neonatal sepsis will fill the gap in the knowledge on the physiopathology of this life-threatening disease but will also offer new alternative targets for therapeutic interventions [8]. Strategies focusing on VFs rather than bacterial survival/growth will apply milder evolutionary pressure for resistance development as most virulence tracts are not essential for survival. Moving in this direction, the implementation over the last decade of next-generation sequences data analysis shows promise in deepening the understanding of VFs associated with GNB pathogenetic strains using a hypothesis-free approach rather than analysing putative virulence-genes by PCR. The general aim of this review was to assess the role of GNB VFs in neonatal sepsis. We primarily aimed to investigate the main VFs leading to adverse outcome in clinical studies and secondarily to evaluate VFs associated with increased invasiveness/pathogenicity in neonates.

## 2. Materials and Methods

Medline (Ovid MEDLINE(R) 1946 to March Week 2 2022), Embase (Embase 1974 to 2022 Week 12), and Cochrane Library (Issue 3 of 12, March 2022) databases were systematically searched on 22 March 2022, with a strategy combining MeSH and free text terms for “neonate” and “bloodstream infection” and “Gram Negative bacteria” and “virulence factors”. The full strategy is available as Appendix A. Studies were considered eligible for inclusion if reporting data on the role of virulome/VFs in BSIs caused by *Enterobacterales* among neonates and infants aged 0–90 days.

All relevant studies published from the inception to March 2022 were considered. No language restriction was applied. Papers reporting not primary data were excluded. Publications reporting data on adults or children were included only if data on infants <90 days were reported separately. Two authors (LB and GR) independently reviewed and extracted the data. Disagreements were resolved by discussion with a third author (LF).

The primary outcome was to investigate the pathogens’ VFs with impacts on neonatal mortality. As a secondary outcome, we investigated the association between virulence determinants in GNB isolates and invasiveness/pathogenic potential in clinical settings.

Data on studies’ characteristics and analysis, pathogen characterizations, microbiological methods, clinical features of included neonates, and clinical outcomes were collected when available. To synthetize the results, the publications included were grouped based on population and comparison. Effects across the studies were heterogeneously expressed as odds ratios, direction of effect, or *p* values. Due to this, the wide diversity in VFs investigated in each study and methodological discrepancies in VFs detection, a meta-analysis of effect estimates could not be undertaken. Individual studies’ results were summarized in tables. Evidence of any effect was expressed by means of vote counting based on the direction effect when appropriate.

This review complies with the Synthesis without meta-analysis (SWiM) reporting guideline [9]. Moreover, to assess the quality of reporting of the included studies, the Strengthening the Reporting of Observational Studies in Epidemiology for Newborn Infection (STROBE-NI) statement was used [10]. The proportion of STROBE-NI items adequately reported was calculated for each study

## 3. Results

### 3.1. Study Selection and Description

In total, the search identified 1243 studies. Among them, 18 papers and 2 abstracts fulfilled our inclusion criteria and were included in the final analysis. Three hundred thirty studies studies were excluded based on the title, 551 were rejected on abstract, and 342 were rejected on full text (Figure 1).

Among the included papers, only four studies evaluated the association between VFs in GNB sepsis and neonatal mortality (primary endpoint). Overall, 16 publications were deemed eligible for the secondary endpoint as they evaluated in clinical settings the association between presence of VFs in GNB isolates and invasiveness/pathogenic potential and were therefore included in the secondary analyses.

### 3.2. Risk of Bias Assessment

Studies included were mainly observational prospective cohort or cross-sectional studies covering the time period for recruitment from 1977 to 2018. According to the STROBE-NI checklist [10], the included articles adequately reported a mean of only 35.4% (range 8.7–78.3%) of the suggested items (Appendix A). A total of 13 out of 20 studies were designed mainly for microbiological purpose. No significant difference was highlighted in terms of compliance with the checklist between studies primarily designed for clinical and those mainly for microbiological purposes (36% vs. 35%, *p* > 0.92).

### 3.3. VFs in GNB Isolates and Neonatal Mortality

The main results are summarized in Table 1. Four studies reported the association between bacterial VFs and neonatal mortality. One focused the analysis on sepsis due to *E. coli* [11], one on *Klebsiella pneumoniae* [5], the third analysed both *E. coli* and *K. pneumoniae* strains [12], and the last included a wider group of GNB (*E. coli*, *Klebsiella* spp., *Enterobacter* spp., *Serratia* spp./*Proteus mirabilis*) [13].

Three studies used a genome-wide approach to investigate the impacts of a pathogen’s determinants on neonatal sepsis morbidity and mortality [5,12,13]. Studies investigating the association between number of virulence genes (VGs) and neonatal mortality showed opposite results. Folgori et al. reported that among different isolates, the number of classes of VGs carried per each isolate was significantly associated with neonatal mortality in the univariate regression [13]. On the contrary, Thomson et al. reported that a lower virulence factor score for *E. coli* was associated with mortality, while no associations were found for *K. pneumoniae* [12].

No single gene showed a consistent association with mortality due to sepsis among the four studies. One study testing *E. coli* isolates for the presence of 12 VGs by polymerase chain reaction (PCR) reported that only the *hek*/*hra* genes were more prevalent in isolates from newborns who died compared with those from survivors [11]. Sands and colleagues used a genome-wide approach to analyse 258 isolates of *K. pneumoniae* and found that the presence of the acquired virulence determinants as reported in the Kleborate tool/repositoire [14] aerobactin and/or salmochelin, with or without yersiniabactin and without colibactin (Kleborate virulence score 3 and 4), was significantly associated with higher mortality compared with those with none of the acquired virulence or yersiniabactin only [5]. Folgori et al. reported that no single virulence determinant was associated with death. Moreover, since different VFs among different strains can play similar functions during the infection process, they also investigated if any VGs class was associated with death, but no significant result was found [13].

### 3.4. VFs Associated with Invasiveness/Pathogenic Potential

To better understand the role of virulence determinants in neonatal sepsis, we investigated in clinical studies if the presence of any VGs/VFs was associated with a surrogate measure of invasive potential (secondary endpoint). Overall, 16 studies and 2 abstracts were included in this secondary analysis [5,11,12,13,15,16,17,18,19,20,21,22,23,24,25,26,27,28,29,30]. With the exception of a single abstract analysing *K. pneumoniae* isolates, all the studies focused on *E. coli* and used a candidate-gene/factor search approach. To summarise the results, according to the information reported in the included studies, we grouped the publications *a-posteriori* in five main categories, based on the type of population and comparison:Comparison of virulence determinants between strains from neonatal sepsis and strains causing non-invasive disease, commensal strains, and environmental isolates (Table 2);VFs in invasive strains causing bacteraemia in neonates vs. children/adults (Table 3);VFs in strains isolated from EOS vs. LOS (Table 4);Assessment of virulence determinants in isolates from neonatal sepsis depending on the portal of entry (Table 5);VFs in strains from sepsis in term vs. preterm neonates.

#### 3.4.1. Invasive and Non-Invasive Strains

Ten publications tried to investigate the virulence determinants in neonatal sepsis comparing clinical isolates from septic newborns with isolates from commensal flora (intestinal and/or vaginal), environment, and non-bacteraemic infections (mostly pyelonephritis with negative blood cultures) (Table 2).

Four studies reported that VFs were underrepresented in faecal strains and noninvasive strains when compared with isolates from neonatal sepsis [16,18,19,20]. Watt and colleagues performed a comparative analysis of VFs in *E. coli* isolates from intestinal flora, vaginal flora, and amniotic fluid of mothers with chorioamnionitis, asymptomatic neonates, and neonatal sepsis strains (blood and cerebrospinal fluid (CSF)) and found that while intestinal isolates had fewer VGs compared with the vaginal flora, there was little difference in the number of VGs between *E. coli* strains from maternal vaginal flora and strains isolated from the amniotic fluid and blood of septic neonates [19].

Among the single gene candidates, hemolysins and genes involved in adherence to the host cells were more frequently associated with invasive potential. Five studies investigated the potential role of hemolysin. Three of them found that the hemolysins were more frequently carried by invasive strains when compared with faecal flora [15,18,23]. Bonacorsi et al. found that the distribution of hemolysins as individual virulence determinants did not differ significantly between the bacteraemic and non-bacteraemic strains from infants with urinary tract infections (UTIs) [16]. On the contrary, when considered the pair K1 antigen (Ag) and/or the hemolysin gene, which was significantly associated with invasive strains in both the univariate and the multivariate regression [16]. One study also observed that the *hly* gene appeared to be significantly less likely to be harboured in CSF strains from neonatal meningitis compared to strains from the blood culture of neonates with bacteraemia [19].

Only four studies investigated if *neuC*/k1 antigen was disproportionally represented among neonatal sepsis strain and noninvasive strains, and they reached contradictory conclusions: Two studies observed *neuC* more frequently in sepsis isolates compared with faecal flora [15] and non-bacteraemic UTIs (in one, the association was found with the alternative presence of K1 antigen and/or the hemolysin gene) [16], while the other two reported a not differential distribution of K1 between bacteraemic strains in newborns and faecal flora [18,19].

Finally, most of the studies reported that genes involved in the adherence to the host cells were more frequent in invasive strains compared with commensal strains. Specifically, four studies reported a significant association between sepsis strains and S fimbriae when compared with intestinal commensal *E. coli* isolates [15,17,18,19]. Additionally, type 1 fimbriae/*fim* gene were observed more frequently in invasive strains compared with faecal flora [15], in strains from amniotic fluid compared with vaginal and intestinal flora [19], and, as for *hly*, in strains from BC compared with CFS [19].

#### 3.4.2. VFs Associated with Invasive Infection in Neonates and Infants <90 Days vs. Older Children/Adult Bacteraemia

We retrieved five studies that compared strains from invasive infections in neonates vs. older children and one vs. adult bacteraemia, trying to unravel the virulence determinants specific to the neonatal sepsis’ pathophysiology [15,18,25,26,27] (Table 3).

Three publications found that neonatal sepsis’ strains harboured a higher number of virulence determinants when compared with strains from older children, although in one study, the difference was not significant [18,25,26].

Four out of five studies investigated the distribution of *neuC*/k1 antigen between neonates vs. children/adults, showing that K1 Ag was more frequent in neonatal invasive infections [15,18,25,27].

Virulence determinants involved in the iron metabolism were assessed only in two studies [25,26]. One of them reported that the *irp2* and the *fyuA* genes were more frequently found in neonatal isolates [25].

Two studied included in this group also reported a comparison in term of antimicrobial resistance in neonates vs. children invasive infections showing conflicting results [25,26].

#### 3.4.3. VFs in EOS vs. LOS

Five studies evaluated if any difference, in terms of virulence determinants, was present among isolates from EOS (<72 h) vs. LOS [5,11,26,28,29] (Table 4).

In two studies, the average number of VFs was higher in strains isolated from EOS [11,26] One of the studies also reported a decreasing trend in the number of VFs based on the timing of sepsis presentation, with an even higher number of virulence determinants reported when the analysis was restricted to neonates affected by sepsis in the first 24 h of life. However, both results did not reach any statistical significance [26].

None of the associations found between individual bacterial VFs and early- or late-onset infections showed consistency among the studies included. Soto et al. reported that when the virulence background of the two groups was compared, no statistically significant differences were observed with the exception of the *ibeA* gene, which was significantly more prevalent in the strains involved in early infections, and the *papGIII* allele which was significantly more prevalent in the strains involved in late infections [28]. Furthermore, the hemolysins were more frequent in EOS, but the result did not reach statistical significance [28]. The opposite feature was observed by Farah Mahjoub-Messai and colleagues, who reported that the hemolysin *hlyC* was significantly more common for LOS vs. EOS [29]. The same study also reported that the virulence determinants showing disproportional occurrence between EOS and LOS were: salmochelin (*iroN*), plasmidic traits (*cva, etsC, iss, ompTp*, and *hlyF*) and the conserved virulence plasmidic region (CVP), characterized by simultaneous presence of *cvaA, etsC, iss, ompTp, hlyF, iucC, iroN*, and *sitA* [29].

Only one study investigated the differences in virulence backgrounds among *K. pneumoniae* isolates from EOS and LOS. Interestingly, the odds of LOS were 89% lower for infants with sepsis due to *K. pneumoniae* strains who had the virulence pattern characterized by the presence of aerobactin and/or salmochelin, with or without yersiniabactin and without colibactin (Kleborate virulence score 3 and 4), compared with those with no acquired genes [5].

#### 3.4.4. VFs Distribution in Neonatal Sepsis Due to Different Portal of Entry

Both the studies included assessed if the *E. coli* strains causing bacteraemia in young infants exhibit different virulence genotype when the infection was acquired via gut translocation compared to the urinary tract (Table 5) [25,29].

Farah Mahjoub-Messai et al. reported that in terms of number of VGs, the two groups were comparable [29]. They also investigated the distribution of the single genes, showing that *papGII* and *tcpC* were less frequently harboured by isolates from the gut than in those from the urinary tract, whereas the *ibeA* gene was far more prevalent in isolates from the gut than in those from the urinary tract [29]. The second study included by Burdet et al. investigated the determinants of severity in children’s bacteraemia. Although the publication did not focus on neonates, the authors reported that severe neonatal bacteraemia was more frequently associated with the digestive portal of entry. In both the univariate and the multivariate regression, a nonurinary source of bacteraemia was the only risk factor significantly associated with sepsis severity; neither the association with prematurity, the birth weight, nor any of the bacterial characteristics tested as significant [25].

#### 3.4.5. VFs in Strains from Sepsis in Term vs. Preterm Neonates

A single publication by Weissman et al. aiming to characterize the *E. coli* isolates from EOS reported differences between strains causing sepsis among premature (22–36 weeks) vs. term neonates [30]. They found that premature infants were significantly more likely to have strains from phylogroups B2 or D and virulence factors score of 2 or more (defined as the carriage of 2 or more of 5 virulence markers *papA* and/or *papC*, *sfa/foc*, *afa-dra, iutA*, and *kpsMTII*) compared with term infants [30].

## 4. Discussion

This systematic review included 20 publications evaluating the role of VFs in neonatal sepsis due to *Enterobacterales*. Only four studies reported data on associations between pathogen virulence determinants and neonatal mortality following sepsis. We included 16 studies investigating the role of VFs in neonatal sepsis through comparisons of virulence profile between invasive and non-invasive isolates, isolates from sepsis in neonates vs. older infants and children, from EOS and LOS, and finally isolates causing neonatal sepsis through different portal of entry. Great heterogeneity was found in terms of VFs analysed and reported, included population, and microbiological methods, with the included studies often showing conflicting data. This variability hampered the comparison of the results. The quality of reporting assessed by the STROBE-NI statement’s checklist was suboptimal in the great majority of the published studies.

We reported no agreement on whether the carriage of more VFs was associated with fatal outcome. One study, to overcome the fact that different VFs can play similar function and therefore their presence is often alternative in GNB isolates, analysed VGs grouped in functional classes and showed that isolates expressing a greater number of virulence classes were associated with mortality, but the results were not confirmed in the multivariate analysis [13]. In the opposite direction, Thomson and colleagues showed an inverse correlation between number of VFs in Escherichia coli isolates and mortality [12]. However, discrepancies between outcome and microbiological results may be partially due to the number of neonates lost to follow-up or to clinical and environmental factors related to the LMIC setting that can contribute to neonates’ death [12].

No single VG showed consistent association with neonatal mortality. One study on a small sample found that among *E. coli* strains isolated from septicaemic neonates, the *hek/hra* genes were associated with death [11], but no other studies confirmed this finding, pointing in the direction that pathogens responsible for neonatal sepsis are widely heterogenous and do not probably follow a unique way to develop invasive disease. However, these conflicting data can also reflect the fact that neonates represent a heterogeneous population, with host factors also playing a crucial role in determining the outcome. For instance, there is evidence that the *E. coli* isolates causing bacteraemia in compromised children are of lower virulence than other bacteraemic isolates [31]. Similarly, underlying medical illnesses, urinary tract abnormalities, and urinary tract instrumentation independently predicted a decreased requirement for P fimbriae in *E. coli* strains from adults and children urosepsis [32], demonstrating that host conditions may nullify normal anatomic and functional defence mechanisms and allow even less virulent organisms to initiate invasive disease.

*E. coli* strains involved in neonatal infections likely originate from the commensal natural flora of pregnant women, which have adapted to survive various ecological media (vaginal, amniotic fluid, blood, etc.), multiplies, and disseminates in the host. Physiological conditions of different niches impose a selection on bacteria leading to progressive change in the *E. coli* subpopulation. The studies included agreed with the fact that the ability of some *E. coli* strains to invade the blood of neonates correlates with an enrichment in number of VFs when compared to commensal strains [16,18,19,20]. Interestingly, one study pointed out that while a difference in the number of VFs could be found between faecal and invasive strains, no differences were reported comparing vaginal strains and invasive ones, suggesting that the population change is first imposed by the mother’s vaginal ecosystem [19]. Analysis at the individual gene level showed that isolates from septicaemic newborns were more likely to harbour *hly* gene and adherence genes such as S fimbriae, type I fimbriae and *papC* (type P pili). Heterogenous results were found about *neuC*/k1 capsular antigen distribution between commensal and bacteraemic strains. Of note, Watt et al. did not observe *neuC* more often in bacteraemic strains when compared with commensal ones, but when comparing with meningitis strains, the resulting difference was significant [19]. The K1 capsular antigen has been reported to be responsible for up to 80% of neonatal meningitis [33]. However, its prevalence among isolates from neonatal bacteraemia has been recently questioned [34] and the estimated prevalence among publications here included ranges from 35% to 59% [11,15,18,19,25,27]. VFs relevant for the pathogenesis of neonatal meningitis may be different from those that determine the development of the initial bacteraemia event.

Since GNB bacteraemia in children and adults tends to occur in a debilitated host with altered defence mechanisms or from localized infection, in contrast to spontaneous occurrences of bacteraemia in healthy newborns, differences in the characteristics of the infecting strains between the two might be anticipated. Results summarised in this review pointed out a higher number of VGs in neonatal isolates and tend to agree on a disproportional distribution of *neuC*/k1 antigen, more frequently carried out by neonatal isolates [15,18,25,27]. However, the five studies here included used different definition of the neonatal period (<90 days, 0–21 days, <7 days) therefore hampering the comparison of the results.

We summarised the findings about the different virulence background in strains isolated from EOS and LOS and from bacteraemia due to gut translocation compared with other portals of entry. The common hypothesis under the analysis carried out in these studies is that in the EOS the pathogen is acquired from the mother before or during birth, likely colonizing the host, before entering the digestive tract and spreading to the bloodstream by gut translocation. EOS seems to be caused by strains with a higher number of VFs, even if this finding was not univocal among the studies [11,26]. Among *K. pneumoniae* isolates causing sepsis in LMIC, the presence of virulence-acquired genes correlated with the early onset of the infection and also with fatal outcome, suggesting that these genes may be involved either in a quicker onset of the infection or reflect the transmission of distinct isolates from the mother’s microbiota or from the clinical environment [5]. In fact, recent findings from LMIC are challenging the assumption of attributing the EOS to vertical transmission from the mothers. In a recent publication, a striking similarity between the pathogens’ profile of EOS and LOS was observed, suggesting that the source of infection, also in EOS, might be environmental [35]. No consistent results were observed at a single-gene level in the studies included.

Few studies have been published so far trying to synthetize the role of the VFs repertoire among different pathogens and/or different clinical settings. To the best of our knowledge, there are no publications reviewing the role of virulence determinants in invasive infections caused by *Enterobacterales* among either adults or children with sepsis. The virulence determinants in *E. coli* responsible for urinary tract infections were recently revised by Bunduki et al. who, using data from clinical syndromes, showed that the VFs with the highest prevalence were those responsible for immune suppression, particularly the *shiA* gene, followed by adhesins genes (CSH and *fimH*/MSHA) [36]. Very few studies tried to investigate the role of VFs in GNB infections in experimental models. After the first observation that strains causing meningitis in newborns carried a K1 capsular antigen, the pathogenetic mechanism underlying neonatal meningitis have been investigated in experimental models, and several VFs relevant for the invasion of the blood brain barrier have been characterized including *ompA, ibeA, cnf1* and *neuC*/k1 [37]. The pathophysiology of neonatal sepsis prior or without meningitis has received far less attention. Experimental models of neonatal *E. coli* bacteraemia have proven that septicaemic *E. coli* strains have the ability to translocate through the neonatal intestinal barrier and enter the blood circulation via mesenteric lymph nodes [6]. However, only few VFs have been demonstrated to be relevant in the bacterial interaction with the intestinal epithelium. One of the most promising candidates is the colibactin gene, which has been associated with enhanced translocation across the intestinal epithelium [38,39,40] as well as involved in the intestinal colonization and virulence in newborn animals [41]. The studies included in this review did not show any association between invasive strains and the colibactin gene, although it might be speculated that the presence of genotoxic *pks*-positive *E. coli* among the early colonizers, which will constitute the gut microbiota, can alter the integrity of the intestinal barrier and facilitate the translocation of other strains. If proven so, *pks*-positive *E. coli* could be the target of microbiota-modulating therapies.

Antimicrobial resistance is a complex problem that requires a multisectoral approach involving a better control of factors that facilitate the emergence and spread of resistance as well as the development of new therapeutic agents that operate under different principles to the currently available antibiotics. In this regard, antivirulence therapy has been envisioned as a promising alternative with the aim of controlling the microbial’s virulence in a pathogen and disease specific fashion, without exerting strong selective pressure [42]. The spectrum of anti-virulence compounds is currently focusing on *Pseudomonas aeruginosa, Enterobacterales* spp., *Staphylococcus aureus* and *Clostridioides difficile* with several bacterial targets which showed promising in preclinical studies, including inhibition of quorum sensing, biofilm formation, adhesion, diverse regulators and persisters [43]. The selection of the targeted VFs is of critical importance for the effectiveness of anti-virulence agents which are often pathogen-specific or patient-/disease-specific. Thus, a clear understanding of the virulence underlying mechanisms and how they interact with specific hosts to produce clinical disease is needed to design new agents.

The present review summarized the findings on the role of virulence determinants in neonatal sepsis caused by Gram-negative bacteria focusing on blood isolates. We performed a broad search including all studies from the inception, and since the pathogen characterization is often reported as a secondary analysis, specific names of VFs and virulence classes were added to the search strategy to maximize the recall. However, this review has several limitations. The studies included were widely heterogeneous. The included publications were grouped based on the population included and/or the evaluated comparator in order to provide a better synthesis of the evidence. Moreover, differences in the definition of the neonatal population, the microbiological approaches used to detect VGs, and the evaluated clinical setting hampered the comparison of the results. It is also worth mentioning that we are moving to genome-wide studies performing microbiological characterization of the pathogens on a large number of samples which, while offering a complete characterization on the bacteria, often lack clinical information on the patients included, limiting the ability to understand the host–pathogen interaction to the fullest.

## 5. Conclusions

No consistent association between virulence determinants and GNB strains causing neonatal sepsis was found in the literature. This may indicate that pathogens responsible for neonatal sepsis are widely heterogenous and can use different pathways to develop invasive disease. However, the wide diversity in host factors and methodological discrepancies might alone explain the disparity of conclusions drawn among different studies. Advances in molecular biology, genomics, and bioinformatics have already contributed to the molecular identification and functional analyses of a wide range of microbial VFs and showed promising in the identification of patterns of clinical interest. However, the genome-wide approach calls for different methods that need to include multicentre studies with larger sample sizes, analyses of VFs/VGs profiles instead of single VF genes, alongside a comprehensive collection of clinical information. GNB are becoming an increasingly prevalent cause of neonatal sepsis especially in LMIC, and the parallel rise of antibiotic resistance calls for new treatment strategies. A better understanding of the roles of VGs in neonatal GNB bacteraemia may offer new vaccine targets and new markers of highly virulent strains that are present in the birth canal or that later colonize the babies that can potentially be used to target screening and preventive interventions and finally new targets for antivirulence therapies.

## Figures and Tables

**Figure 1 ijms-23-11930-f001:**
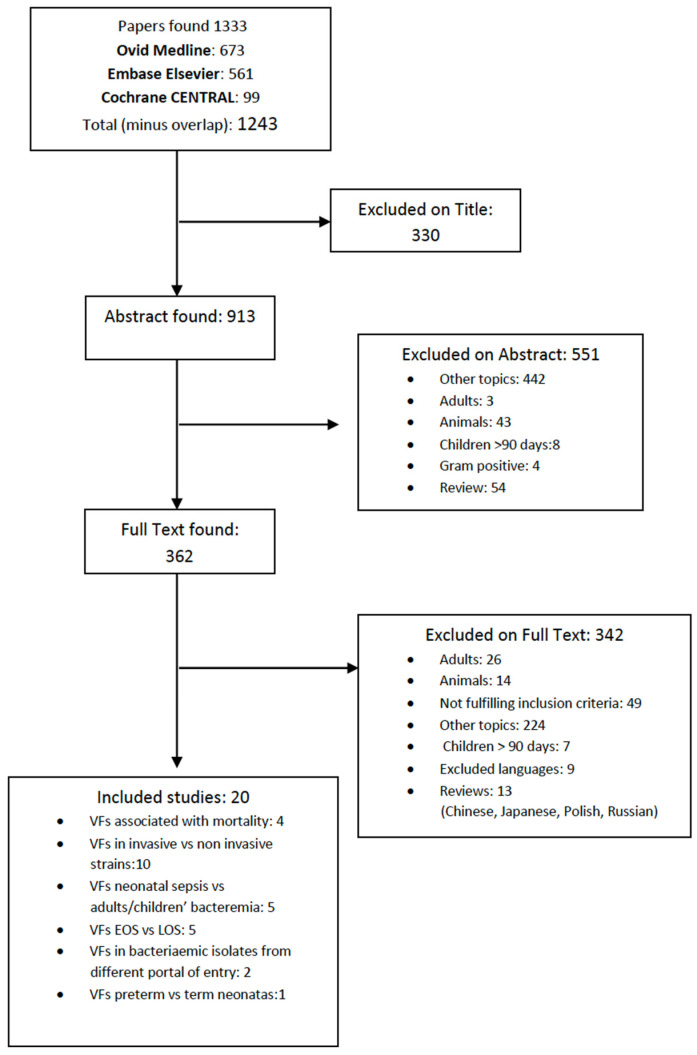
Identification, screening and eligible publications.

**Table 1 ijms-23-11930-t001:** Virulence factors (VFs)/virulence genes (VGs) associated with mortality among septicaemic neonates.

Study	Pathogen/s	Microbiological Method	VFs/VGs	Mortality Rate	Clinical Characteristic	Findings: VF-Outcome	Other Findings
Bryan K. Cole 2019 [10]	*Escherichia coli*	PCR	*cnf1; fimH; hek/hra; hlyC; ibeA; iucC; iro; kpsMT II; nlpI; ompA; papGII-III; sfa/foc*	12/43 (28%)	- Median age at the onset 7 days (IQR 0–10) - Mean gestational age (GA): 32.3 weeks (SD ± 5.4). - 53% history of chorioamnionitis - 40% ET intubated - 42% central venous catheter - 25% Congenital anomalies - 33% developed NEC - 1 meningitis over 14 CFS tested	*hek/hra* more prevalent in isolates from newborns who died compared to those from survivors (*p* < 0.02)	*- cnf1, hek/hra, hlyC, iroN, sfa/focDE* more prevalent in isolates carrying ≥10 VFs - n° of VFs in EOS > LOS (mean ± SD: 8.1 ± 2.4 and 6.8 ± 1.9, *p* < 0.04). - inverse correlation between the total n° of VFs and the n° of non -susceptible antibiotics (*p* < 0.002)
Folgori L. 2021 [12]	*Enterobacterales*	WGS	WGS	18/87 (21%)	- Median age at the onset 15.2 days (IQR 6.7–31) - Median GA of 33 weeks (IQR 28–37) - 56% central venous catheter	N° of classes of VGs per isolates correlated with mortality at 28 days (*p* = 0.002), not significant in Cox multivariate regression	- Positive correlation between the number of resistance and virulence gene per isolate (Spearman’s rank correlation Rho = 0.79; *p* = 0.001)
Kathryn M Thomson, 2021 [11]	*Klebsiella pneumoniae*, *E. coli*	WGS	WGS	-	Not available for sub-group	- Lower virulence factor scores for *E coli* were associated with reported mortality (*U* = 12.50, *p* = 0.042) - No associations between virulence factor score and mortality for *K. pneumoniae* (*U* = 188.0, *p* = 0.66)	- No correlations between pathogenicity Indexing * (PI) and virulence factors scores - No association between PI and outcome for *E. coli* and *K. pneumonie* (*U* = 33.00, *p* = 0.84 and *U* = 178.50, *p* = 0.52)
Kirsty Sands, 2021 [5]	*K. pneumoniae*	WGS	WGS	- Untraceable 43; Dead 58; Alive 157 - Mortality rate 27% (untraceable excluded)	125 EOS 118 LOS 15 ND	OR = 13.77 for mortality following sepsis (*p* < 0.001) among Kleborate virulence score = 3 and 4 ° vs. a virulence score of 0 or 1 ^#^	OR = 0.113 for LOS ( *p* = 0.040) among Kleborate virulence score = 3 or 4 ° vs. virulence score of 0 ^§^

* *bacterial pathogenicity indexes determinate with Galleria mellonella models*; ° virulence score = 3 and 4 ° (presence of aerobactin and/or salmochelin with/without yersiniabactin (without colibactin)); ^§^ virulence score of 0 ^§^ = none of the acquired virulence; ^#^ virulence score of 1 = yersiniabactin only.

**Table 2 ijms-23-11930-t002:** Comparison between bacteraemic strains vs. other strains (non-invasive infections, faecal flora, environmental etc.).

Study	Pathogen/s	Microbiological Method	VFs/VGs	Analysis	Findings
Timo K Korhonen, 1985 [15]	*E. coli*	- Serotyping - Fimbriae analysis: agglutination assays/S fimbriae by colony blotting with anti-S antiserum - Hemolytic activity: sheep blood agar plates	Capsular types, O antigen, Hemolysin, Type 1 fimbriae P fimbriae, S fimbriae, non-P/non -S mannose resistant (MR) adhesins	Comparison VFs in isolates from: - neonatal sepsis in <22 d (10/45 also having meningitis) - infants (22 d-5 m) with BSI (4/18 with meningitis, 2/18 with pneumonia; 2/18 with UTI) - children <14 y with pyelonephritis - feces from healthy donors < 8	- K1 antigen more frequent in neonatal sepsis (67%) than in fecal isolates (22%) and children UTI (31%) - Hemolysin more common in strains from children UTI (60%) than in neonates (18%), and fecal strains (10%) - P fimbriae more frequent in children UTI (76%) vs. neonatal sepsis (38%) - S fimbriae more frequent among neonatal strains (29%) compared with 7% in strains from children UTI and 4% in fecal strains.
Stéphane Bonacorsi, 2005 [16]	*E. coli*	PCR	*Antigen K1, cnf1, hly, papC, papGII, papGIII, sfa/foc, iutA, iroN, fyuA*	Comparison of VFs between infants with bacteraemic and non-bacteraemic UTI (patients with UT abnormalities excluded)	- VFs more frequent in bacteraemic than non-bacteraemic strains but none of the differences for a single factor was statistically significant - The pair *hylC* and/or K1 and *hlyC* and/or *iroN* were more frequent in bacteraeminc strains (*p* = 0.005 and *p* = 0.03) - Only *hlyC* and/or antigen K1 remained significantly frequent in after adjustment for multiple comparisons (*p* = 0.015)
Emma Saez-Lòpez, BS, 2017 [17]	*E. coli* (k1 positive outbreak strain)	RNA-seq	*hlyA, cnf1, sat1, P-fimbriae (papA, −EF, −C), fimH, focG, S-fimbriae (sfa/foc and sfaS), fyuA, iutA, iroN, iucC, ibeA, ibeC, aslA, traJ, hra*	Comparison of the gene expression profile btw K1 positive outbreak strain (N38) and non-outbreak strains	- The outbreak strain carried s-fimbriae (*sfa/foc* and *sfaS*) and PAI-II like domain not present in other K1 strains *- rfaI* and *rfaL* (LPS biosynthesis) and *papI1* were overexpressed in the outbreak strain
Anja Siitonen, 1993 [18]	*E. coli*	- Serotyping: O grouping by agglutination k type by cetrimide reaction+ latex agglutination/immunoelectrophoresis - Fimbriae analysis: agglutination assays/immunofluorescenze - Hemolytic activity: sheep blood agar plates	O antigens, K antigens, H antigens, P fimbriae, S fimbriae, type IC frimbriae, non-P/non -S mannose resistant (MR) adhesins, hemolysins production	-Characterization of O:k:H phenotype, adhesins and hemolytic activity among *E.coli* strain from - neonatal invasive infection(79% non-focal sepsis; 14% meningitis, 7% UTI) - infants’ invasive infection (24% non-focal sepsis; 8% meningitis, 68% UTI - children invasive infection (80% non-focal sepsis, meningitis 5%, UTI, 15%) - faeces	- Strains expressing ≥2 VFs more common among neonatal infection compared to faecal strain (*p* < 0.001) - More frequent in neonatal sepsis compared to facal strains: S-fimbriae (*p* < 0.01) Type 1C fimbriae (*p* < 0.05) Hemolysin (*p* < 0.05) - K1 non differently distributed in strains from neonatal sepsis (49%) vs. faecal flora (33%)
Stephane Watt 2003 [19]	*E. coli*	PCR	*neuC* (K1 capsule antigen), *hly*, *papC*, *sfa/foc, fimH, afa, iucC; ibeA*	Comparison VFs in *E.coli* strains from different anatomical sites: - faecal flora of asymptomatic pregnant women - vaginal flora of asymptomatic pregnant women - amniotic fluid (AF) of asymptomatic neonates born in a chorioamnionitis context - neonates’ blood culture - CSF of neonatal sepsis	- Median N° of VF: intestinal flora 1.5 (range 0–4) vaginal strains 4 (range 0–6). AF 3 (range 1 to 6) neonatal bacteraemia 4 (range 3 to 6) neonatal meningitis 4 (range 1 to 5) *- hly* and *sfa/foc* were more likely isolated in vaginal than intestinal flora (48% vs. 4%, *p* = 0.0006) - Only *fimH* was more frequent in AF strains (96%) than in vaginal strains (76%) and intestinal flora (58%) (*p* = 0.03) - CSF strains compared to fecal strain more likely harbour: *neuC* (83 versus 42%; *p* = 0.04) *iucC* (92 versus 33%; *p* = 0.003) *sfa/foc* (33 versus 4%; *p* = 0.05), *ibeA* (42 versus 4%; *p* = 0.017) - CSF strains compared to blood strains less likely harbour: *hly* (17 versus 65%; *p* = 0.027) *fimH* (67 versus 100%; *p* = 0.041)
R. Tapader, 2014 [20]	*E. coli*	PCR	*- hly, cnf, papC, sfa,* *iroNEC, iucC, ibeA* - SPATEs: *spate, vat, sat, pet, pic, espP, sepA, sigA*	Presence of different VFs and subtypes of SPATEs in neonatal septicemia compared to faecal and environmental isolates	- VFs ≥2 in 39% of isolates from neonatal sepsis vs. 10% in fecal strains and 0% in environmental strain - “SPATE sequence” more frequent in neonatal sepsis strain (89%) in comparison to 7.5% in the faecal isolates (*p* < 0.0001; OR 95% CI = 95.58 (21.03–510.09) and 3% in the environmental isolates [*p* < 0.0001; OR 95% CI = 263.50 (30.42–5926.55)]
A Chmielarczyk, 2013 [21]	*E. coli*	PCR	16 selected VF genes associated with extraintestinal infections (list not available)	VFs in strains from different anatomical sites: - isolates from BSI - isolates from respiratory tract infections (RT) - 8 UTIs	- BSI isolates more frequently carried *ibeA* than isolates from urine and RT (37.5%, 0%, 16% respectively) - UTI isolates carried more frequently *iha* (75% vs. 24% RT vs. 8.3% BSI).
K. Huik, 2010 [22]	*K. pneumoniae*	PCR	*magA* (specific to K1 capsule serotype), *k2A* (specific to K2 capsule serotype), *rmpA* (regulator of mucoid phenotype) and *kfu* (iron uptake system)	VFs in *K.pneumoniae* isolates from neonatal sepsis vs. isolates from surveillance swab (nasopharyngeal -NP and rectal swabs)	- Hypermucoviscosity strains were detected only among mucosal isolates; in 16/30 isolates in pts with and in 0/118 without BSI (*p* < 0.001). - No *magA* and *rmpA* genes were detected. - k2A and *kfu* genes were rare and present in 7/143 and 4/143 mucosal isolates, respectively but only in pts without BSI
Susan W. Cook, 2001 [23]	*E. coli*	PCR	*fim, dra, pap, sfa, foc, uca, papG1, papG3, papG2, hly*	VFs in *E.coli* isolates from women genital tract infections (vaginitis, tubo ovarian abscess) neonatal sepsis and faecal flora	- MRHA+ phenotype (mannose resistant hemagglutination) more frequent in infections isolates than fecal flora (*p* < 0.01) - *Hly*+ more prevalent among tubo-ovarian abscess and neonatal sepsis isolates then faecal isolates (*p* < 0.08) - *Pap* genes were more prevalent among isolates from all three infection sites than fecal flora (*p* < 0.01) - No significant differences in frequency of *fim, sfa, uca (gaf*) or *dra* genes in infection isolates compared with faecal isolates
Subhasree Roy, 2011 [24]	NDM-1 *E. coli*	PCR	*Hly, papaC, sfa, iroN, cnf1, iucC, ibeA*	VFs among NDM-1 *E. coli* isolated from body sites (rectum, groin, mouth) of neonates and blood	- BSI were caused by *E. coli* of phylogroup D - Most VGs (*hly, sfa, iroN, cnf1, iucC, ibeA*) were not found - *PapC* carried in all the bacteraemic strains (4/4) and in 5/16 strains from other body sites

**Table 3 ijms-23-11930-t003:** VF profile in neonatal sepsis vs. older children and adult bacteraemia.

Study	Pathogen/s	Microbiological Method	VFs/VGs	Analyses	Findings
Charles Burdet, 2014 [25]	*E. coli*	PCR	*neuC (K1), iroN, iucC, iha, papC, papGII, papGIII, hlyC, cnf1, hra, sat, ire, usp, ompT ibeA, fyuA, irp2, traT, clbA, clbQ*, *PAI, PAI ICFT073, PAI IIJ96, PAI III536, PAI IV536, PAIgimA, PAIUSP, PAIpks*	Bacterial characteristics in infants ≤ 3 m vs. > 3 m	- N° of VFs higher in ≤3 m of age [median (min-max) = 15 (5–18) vs. 10 (1–19), *p* = 0.02] - VFs more frequent in ≤3 m vs. >3 m: *neuC* (K1 antigen, 55.8% vs. 26.8%, *p* = 0.009), i *irp2* (100% vs. 87.8%, *p* = 0.02) *fyuA* (100% vs. 87.8%, *p* = 0.02) - Severe bacteraemia more frequent in ≤ 3 m (32.6% vs. 7.3% *p* = 0.006) - Antimicrobial resistance scores lower in ≤3 m [median (min-max) = 1 (0–4) vs. 1 (0–5), *p* = 0.01]
Noemí Palma, 2016 [26]	*E. coli*	PCR	*sepA, sigA, pet, pic, espC, hly, cnf1, sat, set1A, set1B, sen, astA, fimA, papC, papGIII, aafC, agg3C, agg4C, iucD, iutA, fyuA shf, agn43, aap, aatA*	Characterization of virulence profile of isolates from infants ≤3 months vs. >3 months	- N° of VGs slightly higher in infants <3 m (5.6 vs. 4.7) * not significant - ESBL more frequent among infants <3 m (*p* = 0.0776) * not significant
Anja Siitonen, 1993 [18]	*E. coli*	- Serotyping: grouping by agglutination k type by cetrimide reaction+ latex agglutination/immunoelectrophoresis - Fimbriae analysis: agglutination assays/immunofluorescenze - Hemolytic activity: sheep blood agar plates	O antigens, K antigens, H antigens, P fimbriae, S fimbriae, type IC fimbriae, non-P/non -S mannose resistant (MR) adhesins, hemolysins production	Characterization of O:k:H phenotype, adhesins and hemolytic activity among E.coli strain from: - neonatal invasive infections (79% non-focal sepsis; 14% meningitis, 7% UTI) - infants’ invasive infection (24% non-focal sepsis; 8% meningitis, 68% UTI - children invasive infection (80% non-focal sepsis, meningitis 5%, UTI, 15%) - faeces	- Strains expressing ≥2 VFs more common among neonatal infection vs. infants with urinary bacteraemia (*p* < 0.05) - K1 more frequently found among neonatal sepsis (49%) compared to infants with UT bacteremia and all invasive strains from children (*p* < 0.05) - P-fimbriae more frequent in infants with UT bacteremia compared to infants with meningitis and nn sepsis (*p* < 0.001) - S-fimbriae more frequent in neonatal strains compared to infants with meningitis and infants with UT bacteremia (*p* < 0.05 and *p* < 0.01)
Timo K Korhonen, 1985 [15]	*E. coli*	- Serotyping - Fimbriae analysis: agglutination assays/S fimbriae by colony blotting with anti-S antiserum - Hemolytic activity: sheep blood agar plates	Capsular types, O antigen, Hemolysin, Type 1 fimbriae, P fimbriae, S fimbriae, non-P/non -S mannose resistant (MR) adhesins	Comparison VFs in isolates from: - neonatal sepsis in <22 d (10/45 also having meningitis) - infants (22 d-5 m) with BSI (4/18 with meningitis, 2/18 with pneumonia; 2/18 with UTI) - children <14 y with pyelonephritis - faeces from healthy donors <8 y	- K1 antigen more frequent in neonatal infection (67%) vs. older infants with invasive infection (22%) *p* < 0.001) - K1 prevalence among meningitis 78% - Hemolysin more common in infants vs. neonates (44% vs. 18%) - No differences in type 1 fimbria distribution according to age or disease
McCabe W.R., 1978 [27]	*E. coli*	Determination of O and K antigen	O antigen, K capsular antigen	Comparison of O and K antigens among *E. Coli* strains from adults and neonatal bacteraemia	KI antigen was significantly greater in neonatal strains compared to adult bacteraemia (*p* < 0.05)

**Table 4 ijms-23-11930-t004:** Comparison VF profile in EOS vs. LOS.

Study	Pathogen/s	Microbiological Method	VFs/VGs	Analyses	Findings: VF-Outcome
S. M. Soto, 2008 [28]	*E. coli*	PCR	*hlyA, cnf1, sat1, papA, papC, papEF, papGII, papGIII, prs 2, fimA, Type 1 fimbria, foc, sfa, fyuA, aer, iucC, iroN, iha, malX* *ibeA, hra*	Comparative analysis of the prevalence of virulence factors in *E. coli* clinical isolates causing EOS vs. LOS	- No differences in phylogenetic groups between EOS and LOS *- ibeA* more prevalent in the strains causing EOS. (86% versus 52%, *p* = 0.01) - *papGIII* more prevalent in the strains involved in LOS (28% versus 5%, *p* = 0.03) *- hlyA* more frequent in the strains causing EOS (41% vs. 20%, *p* = 0.11- not significant)
Bryan K. Cole 2019 [11]	*E. coli*	PCR	*cnf1; fimH; hek/hra; hlyC; ibeA; iucC; iro; kpsMT II; nlpI; ompA; papGII-III; sfa/foc, S*	Comparative analysis of number of virulence factors in *E. coli* clinical isolates causing EOS vs. LOS	- N° of VFs greater in isolates from EOS (mean ± SD: 8.1 ± 2.4 and 6.8 ± 1.9, respectively; *p* < 0.04) - ST95 and ST131 more frequent in EOS vs. LOS but not significant
Farah Mahjoub-Messai, 2011 [29]	*E. coli*	PCR	*Antigen K1, tcpC, papGII, papGIII, sfa/foc, hek/hra, ibeA, cnf1, hlyC, sat, clbN/clbB, vat, cdt, iroN, fyuA, iucC, sitA, cvaA, etsC, iss, ompTp, hlyF*, CVP region	Comparison of VFs in isolates from EOS vs. LOS in bacteremia due to gut translocation	*- hlyC* more common in LOS (46% vs. 0%; *p* = 0.01) - *iroN* was more prevalent in EOS (84% vs. 38%; *p* = 0.02). - Plasmidic traits (cvaA, etsC, iss, ompTp, and hlyF) more frequent in EOS (*p* = 0.01; = 0.02; = 0.04; = 0.04; = 0.04 resp) - CVP region more frequent in EOS (61.5% vs. 15.3%; *p* = 0.01).
Noemí Palma, 2016 [26]	*E. coli*	PCR	*sepA, sigA, pet, pic, espC, hly, cnf1, sat, set1A, set1B, sen, astA, fimA, papC, papGIII, aafC, agg3C, agg4C, iucD, iutA, fyuA, shf, agn43, aap, aatA*	Sub-analysis of VFs in isolates causing EOS vs. LOS	Average number of VGs greater in neonates with sepsis onset in the first 24 h (5.9) than in EOS (5.2) and LOS (4.7)
Kirsty Sands, 2021 [5]	*K. pneumoniae*	WGS	WGS	Sub-analysis of VFs in isolates causing EOS vs. LOS	OR = 0.113 of LOS (*p* = 0.040) among Kleborate virulence score = 3 or 4 (presence of aerobactin and/or salmochelin with/without yersiniabactin (without colibactin)) vs. virulence score of 0 = none of the acquired virulence

**Table 5 ijms-23-11930-t005:** VFs distribution among strains causing infection from different portal of entry.

Study	Pathogen/s	Microbiological Method	VFs/VGs	Analysis	Findings
Farah Mahjoub-Messai, 2011 [29]	*E. coli*	PCR	*Antigen K, tcpC, papGII, papGIII, sfa/foc, hek/hra, beA, cnf1, hlyC, sat, clbN/clbB, vat, cdt, iroN, fyuA, iucC, sitA, cvaA, etsC, iss, ompTp, hlyF*, CVP region	Comparison of VFs from young infants with bacteremia due either to UT or to GT	- Virulence score was nearly identical in UTI and GT isolates (UT 9.7 vs. GT 9.4) - *PapGII* and *tcpC* less frequent in GT than UT isolates (56% vs. 78% *p* = 0.16 and 7.6% vs. 28%, *p* = 0.03) *- ibeA* more prevalent in GT vs. UT (27% vs. 2.7%, *p* = 0.002)
Charles Burdet 2014 [25]	*E. coli*	PCR	*neuC (K1), iroN, iucC, iha, papC, papGII, papGIII, hlyC, cnf1, hra, sat, ire, usp, ompT ibeA, fyuA, irp2, traT, clbA, clbQ*, *PAI, PAI ICFT073, PAI IIJ96, PAI III536, PAI IV536, PAIgimA, PAIUSP, PAIpks*	Bacterial characteristics in bacteremic isolates from urinary vs. digestive origin(GT, gut traslocation)	- No difference found in portals of entry (Urinary vs. GT) - Infants ≤ 3 months with severe bacteraemia had a more frequently non-urinary (92.9% vs. 17.2%, *p* < 0.001) but digestive 50.0% vs. 10.3%, *p* < 0.01) portal of entry than those without - A non-urinary source of bacteraemia was the only risk factor associated with severity (OR = 72.0, 95% CI = 7.2–796.9)

## Data Availability

Data sharing not applicable. No new data were created or analysed in this study. Data sharing is not applicable to this article.

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
