# Peer review of "The Role of Virulence Factors in Neonatal Sepsis Caused by Enterobacterales: A Systematic Review"

_ijms, 2022, doi:10.3390/ijms231911930_

Round 1
Reviewer 1 Report
Valuable review on virulence factors in neonatal sepsis caused by Enterobacterales. Strengths are the systematic design and the study selection. Unfortunately, a huge variability was found in terms of VFs analyzed, included populations and microbiological methods among the 20 publications analyzed in this review, however results are interesting. I recommend publication.
Line 80 : I would change " rather" in "rather than"
Line 393: "that" is repeated twice
Author Response
Authors' response to the general comment
We thank the Reviewer for the positive comment. We concur that the variability of the papers included does not allow to draw conclusions on the topic. However, the main purpose of our work is to summarise what is the knowledge so far and hopefully to stimulate collaboration between investigators and future research in the fields.
R1: Line 80 : I would change " rather" in "rather than"
Authors' response
Thank you for the suggestion.
Line 393: "that" is repeated twice
Authors' response
Thank you for the suggestion.
Reviewer 2 Report
Apart from some spelling errors in table 1, I suggest to better synthetize the data in tables, as some of them may be redundant with the text or too detailed, mainly those related to adults, considering the fact that the subject refers to neonates mainly.
Also, I would suggest to replace the term `enterobacteriales` from the title, abstract or text, with the more medically used term `enterobacteriacee`.
Author Response
R2
Apart from some spelling errors in table 1, I suggest to better synthetize the data in tables, as some of them may be redundant with the text or too detailed, mainly those related to adults, considering the fact that the subject refers to neonates mainly.
Authors' response
Thank you for the suggestions. All the tables have been revised accordingly in the revised version of the paper.
Also, I would suggest to replace the term `enterobacteriales` from the title, abstract or text, with the more medically used term `enterobacteriacee`
Authors' response
Thank you for the suggestion. Even if we recognise that the term “Enterobacteriaceae” is the more frequently used, the taxonomic status of gram-negative bacteria (GBN) has been recently revised and updated (J. Michael Janda, Clinical Micribiology Rev 2021) . Among the changes made, it has been validated the use of the term “Enterobacterales” to indicate an order of GBN which includes different families; the Enterobacteriaceae but also the Morganellaceae, Yersiniaceae, Erwiniaceae, Hafniaceae and others.
The vast majority of the GBN included in our SR are indeed Enterobacteriaceae (E. coli, K pneumonia), however we also included papers analysing pathogens belonging to the Enterobacterales order but from other families such as Serratia (Yersinaceae) and Proteus (Morganellaceae). Moreover, the term “Enterobacterales” but also “Proteus” and “Serratia” (not belonging to the Enterobacteriaceae family) have been used in the search strategy as free text and MESH. For all the above, we hope the Reviewer would concur with us that the use of “Enterobacterales” is the more correct in this case.